# Asymptomatic hypoglycemia among preterm newborns: A cross-sectional analysis

**Shani S. Salum**[1,2,3,4☯], **Florence S. Kalabamu**[1], **Maulidi R. Fataki**[1], **Salha A. Omary**[1,3,4,5], **Ummulkheir H. Mohammed**[1,3,4,6], **Hillary A. Kizwi**[7], **Kelvin M. Leshabari**[3,4,8☯]*

1 Dept of Paediatrics/Child Health, Hubert Kairuki Memorial University, Dar es Salaam, Tanzania, 2 Dept of Paediatrics/Child Health, Muhimbili National Hospital (Mloganzila), Dar es Salaam, Tanzania, 3 Ageing Initiative in Sub-Saharan Africa (AISA) Research Group, Registered Trustees of Ultimate Family Healthcare, Dar es Salaam, Tanzania, 4 Neonatal Network (NeoN), Registered Trustees of Ultimate Family Healthcare, Dar es Salaam, Tanzania, 5 Dept of Paediatrics/Child Health, Temeke Regional Referral Hospital, Dar es Salaam, Tanzania, 6 Dept of Paediatrics/Child Health, Msambweni County Referral Hospital, Msambweni-Kwale, Kenya, 7 Dept of Obstetrics/Gynaecology, Muhimbili National Hospital (Mloganzila), Dar es Salaam, Tanzania, 8 H₃ Clinical Research Unit, I-Katch Technology Ltd, Dar es Salaam, Tanzania

☯ These authors contributed equally to this work.
* celsius_lx@yahoo.co.uk, kelvin.leshabari@ufht.or.tz

**Data Availability Statement:** All relevant data are within the manuscript and its Supporting Information files.

## Abstract

### Background

Hypoglycemia is the commonest metabolic abnormality encountered in newborns. Besides, there is a growing body of evidence that links the causes of early neonatal mortality to neonatal hypoglycemia in Tanzania. However exact factors associated with asymptomatic hypoglycemia in preterm newborns are not known.

### Objective

To assess factors associated with asymptomatic hypoglycemia among preterm newborns.

### Materials and methods

A cross sectional, analytical hospital- based study was carried out at Dar es salaam public regional referral hospitals. Preterm newborns with asymptomatic hypoglycemia were the target population. Data on demographic and clinical characteristics of preterm newborns and their mothers were collected and analyzed using Epi-Info™ software version 7.4. Main data analysis was done by applying a multivariable binary logistic regression model with neonatal random glycaemia coded in a binary fashion at a cut-off point of 2.6 mmol/L. An α-level of 5% was used as a limit of type I error.

### Results

We recruited and analysed 217 preterm newborns within 6–24 hours post-delivery. Male: Female = 1.1:1 (females n = 105, 48.4%). Median glycemic level was 2.6 (IQR; 2.1–3.9) mmol/L. Median gestational age at delivery was 33 (IQR: 30–35) weeks. Breastfeeding within 1st hour post-delivery was a statistically significant factor against glycemic levels

**Funding:** The author(s) received no specific funding for this work.

**Competing interests:** The authors have declared that no competing interests exist.

associated with hypoglycemia (OR; 0.123, 95%-CI; 0.052–0.287) in a fitted multivariable logistic regression model.

## Conclusion

About half of all preterm newborns studied had glycemic values in a statistical range associated with hypoglycemia. Exclusive breast feeding within 1st hour post-delivery was associated with glycemic levels protective from risk of asymptomatic neonatal hypoglycemia.

## Recommendations

Exclusive breastfeeding practices within 1st hour post-delivery may need to be emphasized to all expectant mothers in order to avoid potential risk of asymptomatic hypoglycemia in preterm newborns.

## Introduction

Hypoglycemia is one of the most prevalent metabolic abnormalities encountered in preterm newborn babies [1]. However, the natural history of hypoglycemia in newborns is still a matter of controversy to date. Controversies have both quantitative as well as qualitative origin. For instance, the numerical glycaemic cut-off point of less than 2.6 mmol/L (in US– 50mg/dL equivalent to 2.75mmol/L) still raises concerns to practicing neonatologists the world over [2]. Globally, those on quantitative reasoning do consider the *magic number* (2.6 mmol/L) as the cornerstone of measurements and standardization. However, on evolutionary grounds, practicality of the set *magic number* for diagnosing *neonatal hypoglycaemia* is questionable. Scholars antagonising the concept, base their argument primarily on possible evolutionary advantages to the phenomenon. Those scholars justified their thinking on nosology; based on symptoms and signs (clinical aspects), since for that particular condition, quite often clinicians cannot differentiate physiology from possible pathology using figures alone. Thus, on the basis of qualitative aspects, it is still unclear whether the agreed cut-off point for neonatal hypoglycaemia has a universal clinical utility in neonatology [2].

Prolonged *neonatal hypoglycemia* has been associated with mental retardation, neurological deficits and recurrent seizures. [3] Some studies have shown perinatal hypoxia, small for gestational age and maternal hypertension to be factors associated with hypoglycemia in preterm newborns [4, 5]. However, the exact mechanisms behind the observed negative effects are yet to be identified by neonatologists/endocrinologists to date. Yet still, it is not clear whether the suggested risks are generalized or specific to a certain neonatal cohort alone. At present, there is no evidence from recently available/retrievable published databases, of any findings; on factors associated with neonatal hypoglycemia among preterm babies in Tanzania. Besides, even for findings published worldwide before, almost none specifically assessed the concept of hypoglycemia among asymptomatic preterm newborns.

## Methods

A cross sectional hospital analytical-based study was carried out at Dar es Salaam public regional referral hospitals. Specifically, the study was done at Amana, Mwananyamala and Temeke regional referral hospitals. The three hospitals are the registered public regional referral facilities for Ilala, Kinondoni and Temeke municipalities in Dar es Salaam, Tanzania. We

believed the hospitals to be receiving a representative sample of Dar es Salaam residents. Details about the study settings and design have been documented before [6].

Explicitly, we conceived and recruited all preterm newborns (<37 weeks of gestation) found at neonatal wards and Neonatal Intensive Care Unit (NICU) within 6–24 hours of postnatal life at the three hospitals during the study period. Specifically, the study commenced in June 2022, and run up to (and including) November 2022. A minimum sample of 198 preterm neonates were required in order to achieve a study power of 80% under the assumption of 5% α-level for disproving type 1 error rates in findings. *Preterm newborns* with *asymptomatic hypoglycemia* were the target population. Newborns with either congenital malformations, referred from lower facilities due to other neonatal complications, home delivered or babies symptomatic for hypoglycemia were excluded. Data were collected using a structured questionnaire, after receipt of written informed consent from post-partum mothers. Mothers of preterm newborns were interviewed and their immediate past gestational age assessed using Ballard score chart. Nutritional status was assessed using growth charts. Apgar scores were reported at $1^{st}$ and $5^{th}$ minute post-delivery. Random (venous) blood glucose of preterm newborns were screened using Glucoplus blood glucose monitors (Glucoplus™ Inc. 2004, Canada). Principal Investigator and trained research assistants (neonatal ward nurses) were involved in data collection using the tools (Questionnaire, Ballard score chart, growth chart, Glucoplus™) Both principal investigator and nurses underwent formal pilot practice for 1 day prior to actual use of the tool.

Immediately following data collection, data cleaning followed at the end of each day. It mainly involved checking for consistency in responses given, screening for recording errors as well as coding-decoding of questionnaire responses. Data were thereafter triple entered into the pre-designed template under supervision of study investigators. Data were then stored by principal investigator until analysis time. Data were analysed using Epi-Info™ statistical software version 7.4 (CDC, Atlanta—USA). Descriptive statistics were summarized as median with corresponding inter-quartile range (for quantitative variables) or frequency and proportion (for qualitative variables). Initial data analysis involved exploratory data analysis and mainly screened data for possible linearity/homoscedasticity/normality/non-autocorrelation assumptions between dependent (asymptomatic hypoglycemia) and independent variables (series of clinical and demographic risk factors), outlier analysis, box plots and scatter plots as well as logistic function between outcome variable (hypoglycemia–coded as 1- if random glucose value was < 2.6 mmol/L and 0 –if random glycaemia was ≥ 2.6 mmol/L). Factors associated with hypoglycemia were assessed using binary multivariable logistic regression model. An alpha level of 5% was used as a limit against type I error in findings.

Ethical approval was sought from Hubert Kairuki Memorial University's Institutional Research Ethical Committee as well as the Regional Referral Hospital Authority, which included the Assistant Executive Director of Amana Regional Referral Hospital, the Medical Officer in Charge of Mwananyamala Regional Referral Hospital, and the Medical Officer in Charge of Temeke Regional Referral Hospital. Written informed consent to mothers included description about the study goal, risks and benefits of inclusion of their newborns into the study as well as awareness raising on the voluntary nature of participation into the study. Besides, a note that participation/withdrawal from the study had minimal influence on the planned care at the sites was also part of the written informed consent form. The study was designed to have minimal risks like pain, bleeding, swelling on prick site. These were controlled by applying pressure compression on the prick site. For those preterm newborns found with low blood glucose level, the study team in collaboration with Team providing care in the ward promptly initiated appropriate treatment by administering intravenous 10% dextrose 2mls/kg bolus followed by intravenous infusion and close follow-up to prevent rebound hypoglycemia as per local hospital protocols for management of neonatal hypoglycaemia.

## Results

We recruited and analysed 217 preterm neonates asymptomatic for hypoglycemia at Dar es Salaam Regional Referral Hospitals from June–Nov 2022. They constituted 100, 67 and 50 preterm newborns from Amana, Temeke and Mwananyamala Regional Referral Hospitals respectively. The baseline characteristics of study participants are as summarised in Table 1 below:

## Discussion

In this study, we found almost half of all preterm babies to have been in quantitative hypoglycemic range. It is a matter of special interest to realize that this estimate was derived out of asymptomatic preterm babies. In view of that fact, the common understanding that neonatal hypoglycemia needs to be intervened clinically only upon symptoms need an additional thought to decision makers in health. That assumption may even carry extra weight to neonatologists in resource limited settings. Specifically, it appears preterm delivered babies are potential risky group among neonates for development of hypoglycemia. Thus, hospital policies/protocols and national guidelines may consider these findings as potential signals towards including this specific neonatal group for urgent/frequent glycemic screening. The current findings are comparable to others in similar settings done before [7–10] For instance, Sultan and his colleagues found out 73% of all babies born in South-Eastern Tanzania to had been in hypoglycemic range [7]. However, the study included all neonates (irrespective of gestation age) and is about two decades old. Otherwise, in the same East African zone, there have been some other notable differences in findings to ours. For instance, Mukunya and others in their Uganda's community-based study found a prevalence of 2.2% for neonatal hypoglycemia in their screening exercise [11]. However, Mukunya's study was a secondary analysis to a primary study with different objectives that was designed at a community level [11]. In general, the fact

**Table 1. Baseline characteristics of selected maternal, neonatal and institutional factors associated with asymptomatic hypoglycemia among preterm newborns seen in Dar es salaam public regional referral hospitals.**

| Continuous variables | Median | IQR |
|---|---|---|
| Neonatal age (hours) | 10 | 7–16 |
| Maternal age (years) | 26 | 23–30 |
| Neonatal glycemic level (mmol/L) | 2.6 | 2.1–3.9 |
| Active labour duration (hrs:mm) | 5:29 | 4:15–7:00 |
| Gestational age (weeks) | 33 | 30–35 |
| Birth weight (g) | 1500 | 1200–1900 |
| **Categorical variables:** | **Frequency (n)** | **Percent (%)** |
| **Neonatal sex** | | |
| Female | 105 | 48.4 |
| Male | 112 | 51.6 |
| **Mode of delivery** | | |
| SVD | 183 | 84.4 |
| Assisted delivery | 2 | 1.0 |
| Cesarean Section | 32 | 14.7 |
| **Nutritional status** | | |
| Small for Gestational Age | 36 | 16.6 |
| Appropriate for Gestational Age | 172 | 79.3 |
| Large for Gestational Age | 9 | 4.1 |
| **Breastfeeding within 1st hr post-delivery** | | |
| Yes | 54 | 24.9 |
| No | 163 | 75.1 |

that the metabolic abnormality seems prevalent in Africa, and especially Tanzania, is a wake-up call for urgent policy and guidelines changes on the construct of neonatal hypoglycemia.

Accordingly, the median random blood glucose of 2.6mmol/l among our study population are likely to be justifiable evidence for glycemic screening even among asymptomatic newborns. There are several mechanisms that accounts for possibilities of the metabolic abnormality in neonates. They include limited glycogen and fat stores, inability to generate new glucose using gluconeogenesis pathways, with higher metabolic demands due to a relatively larger brain size, and hyperinsulinemia [11]. However, our study was cross-sectional by design, and hence we could not make follow-up data of the glycemic ranges in studied children nor could we justify the single glycemic output to be significant to justify neonatal hypoglycemia in a clinical sense. However, given our study findings, we can safely assume that glycemic screening need to be instituted to all neonates regardless of their initial clinical stata.

We also tested a number of known associated factors for neonatal hypoglycemia. It was evident from Table 2 that breastfeeding within 1st hour of birth to be the only clinically

**Table 2. Univariate and multivariable binary logistic regression analysis of selected clinical factors associated with asymptomatic hypoglycemia among preterm newborns seen at Dar es salaam public regional referral hospitals.**

| VARIABLE | n (%) | O.R (95% CI) UNADJUSTED | O.R (95% C.I.) ADJUSTED |
|---|---|---|---|
| **Neonatal age (hours)** | | | |
| 6–12 | 132 (60.8) | 1.208 (0.691–2.112) | 1.03 (0.49–4.33) |
| >12 | 85 (39.2) | Ref | Ref |
| **Neonatal sex** | | | |
| Female | 105 (48.4) | 0.662 (0.387–1.134) | 0.52 (0.41–1.87) |
| Male | 112 (51.6) | Ref | Ref |
| **Mode of delivery** | | | |
| SVD | 183 (84.3) | Ref | Ref |
| Ass. VD | 2 (0.9) | 1.286 (0.079–20.874) | 1.00 (0.06–37.1) |
| CS | 32 (14.7) | 1.457 (0.686–3. 094) | 1.21 (0.54–5.5) |
| **Gestation age (weeks)** | | | |
| <28 | 68 (31.3) | 0.845 (0.470–1.518) | 0.67 (0.3–1.51) |
| 28 - <32 | 8 (3.7) | 3.269 (0.662–16.151) | 1.6 (0.43–23.65) |
| 32–36 | 141 (65) | Ref | Ref |
| **1st Apgar score** | 217 (100) | 1.221(0.945–1.578) | 1.13 (0.89–1.61) |
| **5th Apgar score** | 217(100) | 1.297(0.998–1.685) | 1.06 (0.83–2.07) |
| **Nutritional status** | | | |
| SGA | 36 (16.6) | 3.794(0.811–17.745) | 2.87 (0.76–28.1) |
| AGA | 172(79.4) | 1.374(0.335–5.645) | 1.26 (0.32–6.66) |
| LGA | 9 (4.1) | Ref | Ref |
| **Respiratory Distress Synd** | | | |
| Yes | 45 (20.7) | **2.115(1.083–4.128)** | 1.363(0.659–2.818) |
| No | 172 (79.2) | Ref | Ref |
| **Birth Asphyxia** | | | |
| Yes | 13 (6) | 2.785 (0.838–9.254) | 1.93 (0.77–22.2) |
| No | 204 (94) | Ref | Ref |
| **MAS** | | | |
| Yes | 2 (0.9) | 1.216(0.075–19.691) | 1.01 (0.00–30.24) |
| No | 215(99.1) | Ref | Ref |
| **Neonatal Sepsis** | | | |
| Yes | 13 (6) | 1.447(0.470–4.456) | 1.33 (0.39–5.63) |

*(Continued)*

**Table 2.** (Continued)

| VARIABLE | n (%) | O.R (95% CI) UNADJUSTED | O.R (95% C.I.) ADJUSTED |
|---|---|---|---|
| No | 204(94) | Ref | Ref |
| **Radiant/Incubator Use** | | | |
| Yes | 207(95.4) | 1.888(0.481–7.414) | 1.24 (0.64–3.2) |
| No | 10(4.6) | Ref | Ref |
| **Breastfeeding within 1ˢᵗ hour of birth** | | | |
| Yes | 54(24.9) | **0.110(0.048–0.254)** | **0.123(0.052–0.287)** |
| No | 163(75.1) | Ref | Ref |
| **Maternal age** | 217(100) | 1.024(0.971–1.080) | 1.00 (0.99–1.03) |
| **Preeclampsia/Eclampsia** | | | |
| Yes | 30(13.8) | **2.713(1.207–6.098)** | 1.480(0.630–3.482) |
| No | 187 (86.2) | Ref | Ref |
| **Dextrose infusion** | | | |
| Yes | 5(2.3) | 1.837(0.301–11.203) | 1.31 (0.4–48.6) |
| No | 212(97.7) | Ref | Ref |
| **Maternal Diabetes Status** | | | |
| Yes | 7(3.2) | 2.962(0.573–15.305) | 3.78 (0.41–67.2) |
| No | 210(96.8) | Ref | Ref |
| **Maternal Pre-existing medical/surgical conditions** | | | |
| Yes | 27(12.4) | 1.608(0.714–3.619) | 1.1 (0.87–5.97) |
| No | 190(87.6) | Ref | Ref |
| **Type of pregnancy** | | | |
| Multiple | 49(22.6) | 1.354(0.715–2.561) | 1.32 (0.65–3.2) |
| Singleton | 168(77.4) | Ref | Ref |
| **Active 2ⁿᵈ stage of labour duration (hrs)** | | | |
| <12 | 209 (97.3) | Ref | Ref |
| ≥12 | 8(2.7) | 1.223(0.298–5.02) | 1.04 (0.00–7.89) |
| **Delivery time (hrs)** | | | |
| Daytime (0800–1859) | 80(36.9) | 1.359(0.781–2.364) | 0.97 (0.92–1.00) |
| Nightime (1900–0759) | 137(63.1) | Ref | Ref |
| **Neonatal Birth Weight(g)** | | | |
| LBW | 120(55.3) | Ref | Ref |
| VLBW | 92(42.4) | 3.269(0.662–16.151) | 2.57 (0.55–20.13) |
| ELBW | 5(2.3) | 0.845(0.470–1.518) | 1.11 (0.38–1.9) |

NOTE

OR = Odds Ratio, C.I. = Confidence Interval, SVD = Spontaneous Vertex Delivery, CS = Cesarian Section, Ass. VD = Assisted Vaginal Delivery, SGA = Small for Gestation Age, AGA = Appropriate for Gestation Age, LGA = Large for Gestational Age, RDS = Respiratory Distress Syndrome, BA = Birth Asphyxia, NS = Neonatal Sepsis, MAS = Meconeum Aspiration Syndrome, BF = Breast Feeding, LBW = Low Birth weight, VLBW = Very Low Birth Weight, ELBW = Extreme Low Birth Weight.

meaningful factor in preventing asymptomatic neonatal hypoglycemia. Similarly, Mukunya and others in their Uganda's community-based study found that delayed breastfeeding initiation was contributing neonatal hypoglycemia [11]. Breastfeeding has been evidently associated with glucose stability in the 1ˢᵗ 24 hours of life on earth previously among term infants [12, 13]. Otherwise, our findings are consistent and reflective of what was seen before in a large quasi-experimental study from Denmark [14]. Accordingly, the Danish study assessed the contribution of early breastfeeding to prevention of neonatal hypoglycemia using skin-to-skin contact un-interruptive for at least 2-hours immediately post-delivery [14].

Our study findings are unique as they report asymptomatic neonatal hypoglycemia and through analysis, we managed to confer potential solution to the clinical challenge. Moreover, our study findings included maternal, institutional as well as neonatal factors known to be associated with hypoglycemia in asymptomatic preterm babies. Thus, we believe our findings can provisionally be used as a palpable evidence while waiting for more prospective designed evidence in future. However, our findings were limited in time (6–24 hours) by design. The decision to limit the time is subject to earlier observations on the probable physiologic basis of glucose homeostasis early in extra-uterine life [15–17]. Thus, should there be any deviations in glycemic state over time, our study was limited to capture such an information.

On a strict evolutionary sense, there appears to be potential utility of low blood sugar levels to newborns immediately upon entry to extra-uterine life [18]. To what extent do these findings translate to prevalent double burden of morbidity and mortality indices later among under-five year population in Tanzania and beyond, is still a matter of speculation [19–26]. However, the fact that there is solid evidence for under-fives population to be in ill-state in Tanzania [22] calls for urgent considerations for more analytical, possibly longitudinal designs in ascertaining the long-term effects associated with asymptomatic hypoglycemia in newborns. Besides, these findings call for urgent and reliable estimates of morbidity and mortality statistics of special groups in the population pyramid. At present, there are potential indications that shows certain segments of the Tanzania's population are left out in most reported morbidity and mortality statistics [25, 26]. There are potential clues that chronic ill-states in childhood may result to miserable later life years both in Africa and elsewhere [23, 24, 27]. As we are advancing medical sciences on ageing cascades, even among humans [19, 24], we wish to consider that as a *treasure hunt* at least for now.

## Supporting information

**S1 Dataset.**
(PDF)

## Author Contributions

**Conceptualization:** Shani S. Salum, Florence S. Kalabamu, Maulidi R. Fataki, Salha A. Omary, Ummulkheir H. Mohammed, Hillary A. Kizwi, Kelvin M. Leshabari.

**Data curation:** Shani S. Salum.

**Formal analysis:** Shani S. Salum, Florence S. Kalabamu, Maulidi R. Fataki, Ummulkheir H. Mohammed, Hillary A. Kizwi, Kelvin M. Leshabari.

**Funding acquisition:** Shani S. Salum, Hillary A. Kizwi.

**Investigation:** Shani S. Salum, Kelvin M. Leshabari.

**Methodology:** Shani S. Salum, Florence S. Kalabamu, Maulidi R. Fataki, Salha A. Omary, Hillary A. Kizwi, Kelvin M. Leshabari.

**Project administration:** Shani S. Salum, Florence S. Kalabamu, Maulidi R. Fataki.

**Resources:** Shani S. Salum, Hillary A. Kizwi.

**Software:** Shani S. Salum, Hillary A. Kizwi, Kelvin M. Leshabari.

**Supervision:** Florence S. Kalabamu, Maulidi R. Fataki, Kelvin M. Leshabari.

**Validation:** Florence S. Kalabamu, Maulidi R. Fataki, Salha A. Omary, Ummulkheir H. Mohammed, Hillary A. Kizwi, Kelvin M. Leshabari.

**Visualization:** Salha A. Omary, Ummulkheir H. Mohammed, Hillary A. Kizwi, Kelvin M. Leshabari.

**Writing – original draft:** Shani S. Salum, Kelvin M. Leshabari.

**Writing – review & editing:** Shani S. Salum, Florence S. Kalabamu, Maulidi R. Fataki, Salha A. Omary, Ummulkheir H. Mohammed, Hillary A. Kizwi, Kelvin M. Leshabari.

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
