## [Decision Letter · Decision Letter 0]

14 Nov 2023

PONE-D-23-33081Asymptomatic hypoglycemia among preterm newborns: a cross-sectional analysis.PLOS ONE

Dear Dr. Leshabari,

Thank you for submitting your manuscript to PLOS ONE. After careful consideration, we feel that it has merit but does not fully meet PLOS ONE’s publication criteria as it currently stands. Therefore, we invite you to submit a revised version of the manuscript that addresses the points raised during the review process.

We look forward to receiving your revised manuscript.

Kind regards,

Kazumichi Fujioka

Academic Editor

PLOS ONE

Journal Requirements:

4. Please ensure that you include a title page within your main document. You should list all authors and all affiliations as per our author instructions and clearly indicate the corresponding author.

Reviewers' comments:

Reviewer's Responses to Questions

**Comments to the Author**

1. Is the manuscript technically sound, and do the data support the conclusions?

Reviewer #1: Yes

Reviewer #2: Yes

2. Has the statistical analysis been performed appropriately and rigorously? 

Reviewer #1: Yes

Reviewer #2: Yes

3. Have the authors made all data underlying the findings in their manuscript fully available?

Reviewer #1: Yes

Reviewer #2: No

4. Is the manuscript presented in an intelligible fashion and written in standard English?

Reviewer #1: Yes

Reviewer #2: Yes

5. Review Comments to the Author

Reviewer #1: Re: Asymptomatic hypoglycemia among preterm newborns: a cross-sectional analysis.

Thanks for allowing me to review this work. Find below my comments

Short title:

“neonatal hypoglycemia” changes to hypoglycemia in preterm newborns.

Keywords:

“Dar es Salaam” delete, replace with preterm

Abstract

‘There is a growing body of evidence that links the causes of early neonatal mortality to neonatal hypoglycemia in Tanzania’ Replace with another statement, this is a well-established fact globally.

“Written informed consent was sought from each mother prior to inclusion of their babies into the study” delete

“Breastfeeding within 1st hour post-delivery was the only factor significantly associated with neonatal hypoglycemia (OR; 0.123, 95%-CI; 0.052-0.287).” Kindly rephrase to avoid ambiguity. Your odds ratio is less than.

“Nearly half of all asymptomatic newborns were in hypoglycemic range” You can not conclude with findings not in the results section, this is appearing out of no where

“Preterm neonates maybe the risky group worth considering for continuous glycemic screening at hospital protocols” Your abstract failed to support this statement and thus, you can not make this recommendation

Introduction

“They consider their argument on possible evolutionary basis for its occurrence within the first 24-hours of extra-uterine life. They justified their thinking on the nosology to be based on symptoms and signs (clinical aspects),” replace the word “they” and avoid third person pronoun in a scientific writing

“Some studies have shown perinatal hypoxia, small for gestational age and maternal hypertension to be factors associated with hypoglycemia in preterm newborns (4, 5) However, the exact mechanisms behind the observations are yet to be identified by neonatologists/endocrinologists to date. Yet still, it is not clear whether the suggested risks are generalized or specific to a certain neonatal cohort alone” move up to the paragraph before..

“This study gives findings from a cross-sectional sample of preterm neonates who presented with asymptomatic hypoglycemia in typical neonatal set ups” delete

Methods:

“Otherwise, Tanzania is a nation state situated on the East African coast. It is a former German colony, that accidentally fell under British protectorate after the World War II, up to her political independence. Most of her existing infrastructure in health, especially the study settings are post-independence initiatives on health. Tanzania is among the fastest growing nation states in Africa. She is rich in demographic sects, as evidenced by different population sub-structure ranging from Bantus to Afro-Asiatic population groups, especially in Dar es Salaam.” Delete, this is a manuscript and not a dissertation or thesis.

“Thus, the study population was considered to be representative of African demographic sects for all practical and clinical purposes” This is incorrect, kindly delete

“Nutritional status was assessed using growth charts” Mentioned the growth chart name

“Random (venous) blood glucose of the preterm newborns screened using Glucoplus blood glucose monitors (Glucoplus™ Inc. 2004, Canada)” has this been validated among the neonates?

“Data was triple entered under” who do you mean by triple entered? Why?

“(hypoglycemia – coded as 1- if random glucose value was ≥ 2.6 mmol/L and 0 – if random glycaemia was < 2.6 mmol/L).” kindly cross-check, your desired outcome is hypoglycemia and should have been coded 1.

Results:

Apgar score, and weight are appearing here for the first time, how were they assessed and it should be part of the methods. Also, birth asphyxia, respiratory distress, MAS

“TABLE 2: multivariate analysis” it should be bivariate or multivariable binary logistic regression

No mention of how many has hypoglycemia

Discussion

“In this study, we found almost half of all preterm babies to have been in quantitative hypoglycemic ran.” This is not mentioned in your results

”The current findings are comparable to others in similar settings (7-10) For instance, Sultan and his colleagues found out 73% of all babies born prematurely in South-Eastern Tanzania to had been in hypoglycemic range” This statement contradict your earlier assertion that no study has been done in Tanzania (At present, there is no evidence from available/retrievable published databases of any findings on factors associated with neonatal hypoglycemia among preterm babies in Tanzania).

“To what extent do these findings translate to prevalent double burden of morbidity and mortality indices among under-five year population in Tanzania and beyond, is still a matter of speculation (25, 26). However, the fact that there is solid evidence for under-fives population to be in ill-state in Tanzania (22) calls for urgent considerations for more analytical, possibly longitudinal designs in ascertaining the long-term effects ofasymptomatic hypoglycemia in newborns. Besides, these findings call for urgent and reliable estimates of morbidity and mortality statistics of special groups in the population pyramid. At present, there are potential indications that shows certain segments of the Tanzania’s population are left out in most reported morbidity and mortality statistics (25, 26). There are potential clues that chronic ill-states in childhood may result to miserable later life years both in Africa and elsewhere (23, 24). As we are advancing medical sciences on ageing cascades, even among humans (19, 24), we wish to consider that as a treasure hunt at least for now.” Delete, irrelevant.

Reviewer #2: The work is important in it's field. It has a logical flow and would be important in the setting that it was conducted. The authors need to present the work better especially the statistical work for factors associated in Table 2. They also need to discuss their work in more depth other than the simplistic discussion they have presented. There are many factors that are associated with hypoglycemia but these have not been studied at all or even mentioned as limitations like maternal medical conditions like HIV, BMI of mother, use of drugs that could cause hypoglycemia , critical illness; did they only enrol stable infants? Was this also the best design for such a study given the resources they had?

6. PLOS authors have the option to publish the peer review history of their article (what does this mean?). If published, this will include your full peer review and any attached files.

Reviewer #1: **Yes: **Olayinka Ibrahim

Reviewer #2: No

---

## [Author Response · Author response to Decision Letter 0]

18 Jan 2024

Editors/ Reviewer’s comments/suggestions/advice Specific final change made by author Final status (page number)

1. Editor’s advice:

 Upload minimal anonymised dataset Attached as part of this submission Submitted

---

## [Editor Report · Decision Letter 1]

30 Jan 2024

PONE-D-23-33081R1Asymptomatic hypoglycemia among preterm newborns: a cross-sectional analysis.PLOS ONE

Dear Dr. Leshabari,

Thank you for submitting your manuscript to PLOS ONE. After careful consideration, we feel that it has merit but does not fully meet PLOS ONE’s publication criteria as it currently stands. Therefore, we invite you to submit a revised version of the manuscript that addresses the points raised during the review process.

We look forward to receiving your revised manuscript.

Kind regards,

Kazumichi Fujioka

Academic Editor

PLOS ONE

Journal Requirements:

Additional Editor Comments:

Both reviewers requires response comment letter point by point.

PONE-D-23-33081R1

Asymptomatic hypoglycemia among preterm newborns: a cross-sectional analysis.

Dr Kelvin Melkizedeck Leshabari

Dear Editor,

I can not access the comments from the author or the response from the reviewers.

Thank you

PONE-D-23-33081R1

Asymptomatic hypoglycemia among preterm newborns: a cross-sectional analysis.

Dr Kelvin Melkizedeck Leshabari

Dear Editor,

Thank you for asking to review the revised manuscript. I observed that there was no cover letter or point-to-point response to my earlier comments. Besides, I hardly see track changes in the revised manuscript.

I wish these issues to be resolved before proceeding with review.

Best wishes

---

## [Author Response · Author response to Decision Letter 1]

20 Mar 2024

1. Editors’ advice:

Please ensure you include your title page in the main document Title page is part of the manuscript (main) document now. Changed as requested (pp. 1)

2. Reviewer’s advice:

Neonatal hypoglycemia change to ‘hypoglycemia in preterm newborns’ Changed to read as hypoglycemia in preterm newborns. Changed as requested (pp. 1)

3. Reviewer’s advice: 

Keywords – Dar es Salaam, replace it with the word preterm. Changed to preterm Changed as requested (pp. 3)

Reviewer’s advice: 

Abstract section: 

4. “There is a growing body of evidence that links the causes of early neonatal mortality to neonatal hypoglycemia in Tanzania.”

Replace it with another statement, this is a well-established fact globally.

5. “Written informed consent was sought from each mother prior to inclusion of their babies into the study” delete 

6. “Breastfeeding within 1st hour post-delivery was the only factor significantly associated with neonatal hypoglycemia (O.R.: 0.123, 95% C.I.: 0.052 – 0.287)”. Kindly rephrase to avoid ambiguity. Your odds ratio 

Replaced with a totally new sentence of 

“Preterm newborns are the group with the highest risk of clinically significant hypoglycemia”. 

 Deleted.

Changed to

 “Breastfeeding within 1st hour post-delivery was a statistically significant factor against glycemic levels associated with hypoglycemia (OR; 0.123, 95%-CI; 0.052-0.287) in a fitted multivariable logistic regression model”

Changed as requested (pp. 3)

Deleted as requested (pp. 3)

Changed to reflect the appropriate statistical interpretation (refer to results section pp. 3)

Editors’/Reviewers’ comments-advice Specific final change made by authors Final status (page number)

 Reviewer’s comments 

Abstract section (cont…)

7. “Nearly half of all asymptomatic newborns were in hypoglycemic range.” You cannot conclude with findings not in the results section, this is appearing out of nowhere.

8. “Preterm neonates maybe the risky group worth considering for continuous glycemic screening at hospital protocols”. Your abstract failed to support this statement and thus, you cannot make this recommendation. 

Changed to read as 

“About half of all preterm newborns studied had glycemic values in a limit associated with hypoglycemia in newborns”.

Reason: 

Definition of the term ‘median’ as a statistical measure of central tendency. The median random glycaemia among study participants was 2.6 mmol/L (see table 1 in results section pp. 7).

Deleted

 Changed accordingly with retention of the glycemic value of 2.6 mmol/L

Deleted 

Editors’/Reviewers’ comments-advice Specific final change made by authors Final status (page number)

Reviewer’s comments 

Introduction section:

9. “They consider their argument on possible evolutionary basis for its occurrence within the first 24-hours of extra-uterine life. They justified their thinking on the nosology to be based on symptoms and signs (clinical aspects),” replace the word “they” and avoid third person pronoun in a scientific writing.

10. “Some studies have shown perinatal hypoxia, small for gestational age and maternal hypertension to be factors associated with hypoglycemia in preterm newborns (4, 5) However, the exact mechanisms behind the observations are yet to be identified by neonatologists/endocrinologists to date. Yet still, it is not clear whether the suggested risks are generalized or specific to a certain neonatal cohort alone”. 

11. “This study gives findings from a cross-sectional sample of preterm neonates who presented with asymptomatic hypoglycemia in typical neonatal set ups” Delete 

Replaced with words

“Scholars antagonising the concept, base their argument primarily on possible evolutionary advantages to the phenomenon. Those scholars justified their thinking on the nosology, based on symptoms and signs (clinical aspects), since for that particular condition, quite often clinicians cannot differentiate physiology from possible pathology using figures alone”.

 Moved accordingly

Deleted accordingly 

Changed as per reviewer’s suggestions.

Changed as per reviewer’s suggestions.

 (see pp. 4)

Deleted as per reviewer’s recommendations

Editors’/Reviewers’ comments-advice Specific final change made by authors Final status (page number)

 Reviewer’s advice:

 Methods section:

12. “Otherwise, Tanzania is a nation state situated on the East African coast. It is a former German colony, that accidentally fell under British protectorate after the World War II, up to her political independence. Most of her existing infrastructure in health, especially the study settings are post-independence initiatives on health. Tanzania is among the fastest growing nation states in Africa. She is rich in demographic sects, as evidenced by different population sub-structure ranging from Bantus to Afro-Asiatic population groups, especially in Dar es Salaam” Delete, this is a manuscript and not a dissertation or thesis 

13. Thus, the study population was considered to be representative of African demographic sects for all practical and clinical purposes. This is incorrect, kindly delete

14. What was the definition of neonatal hypoglycemia in preterm? 

 Deleted

Deleted

 Neonatal hypoglycemia was defined as a random blood glucose level ≤ 2.6 mmol/L whether or not they presented with symptoms and/or signs consistent with clinically defined hypoglycemia. 

Deleted as per reviewer’s advice 

 (see pp. 5)

Deleted as per reviewer’s recommendations (see pp. 5)

 Unchanged

Editors’/Reviewers’ comments-advice Specific final change made by authors Final status (page number)

 Reviewer’s advice:

 Methods section (cont…):

15. What was the definition of preterm? How did you assess prematurity?? Using LNMP or Ballard score?? 

16. Did you include children who had received IV dextrose infusion If so, how many hours since the infusion?

17. Are the characteristics of the children admitted at these three hospitals the same?

18. “Data was triple entered under” who do you mean by triple entered? Why?

Prematurity was assessed using Ballard score

All children who received IV dextrose infusions were excluded from the study.

For all practical and clinical purposes – yes.

Besides, the study settings (all three hospitals) are for all practical purposes considered to be the same as they are in the same level in the Tanzania’s referral system (i.e. regional referral hospitals). Furthermore, they serve as public facilities with more or less the same population in terms of demographics, socio-economic stata, etc

Data were entered thrice during data entry process by three separate data clerks. 

Reason: To avoid possibilities for data collection and/or data entry errors to the maximum during data collection/storage stage. Multiple data clerks brought data to the entry sites, and there were therefore potential risks for erroneous data inclusion/deletion/misrepresentation.

 Unchanged

Unchanged

 Unchanged

 Unchanged

Editors’/Reviewers’ comments-advice Specific final change made by authors Final status (page number)

 Reviewer’s advice:

 Methods section (cont…):

19. “(hypoglycemia – coded as 1- if random glucose value was ≥ 2.6 mmol/L and 0 – if random glycaemia was < 2.6 mmol/L).” kindly cross-check, your desired outcome is hypoglycemia and should have been coded 1.

Results section 

20. “TABLE 2: multivariate analysis”. It should be bivariate or multivariable binary logistic regression. No mention of how many has hypoglycemia 

Agreed. There was a typo during typing as in reality what was done included considering outcome of interest (coded as 1) whenever random glycaemia was ≤ 2.6 mmol/L and anything else as normoglycemia (coded as 0). The sentence has been rephrased to reflect the correct meaning of what was done during exploratory data analysis and even final data analysis.

Agreed and changed accordingly

Changed accordingly 

(refer pp. 6)

Changed in accordance to reviewer’s recommendations. (see pp. 7)

Editors’/Reviewers’ comments-advice Specific final change made by authors Final status (page number)

 Reviewer’s advice:

Discussion section:

21. “In this study, we found almost half of all preterm babies to have been in quantitative hypoglycemic range”. This is not mentioned in your results.

 Kindly refer to table 1 in the results section under median values of neonatal glucose levels (in mmol/L). Median was the measure of central tendency used during analysis as data was found to be skewed. Besides, the median is statistically defined as the 50th percentile of the values of an item/variable under study. the median value of random glycemia for the studied newborns was 2.6 mmol/L. Standard quantitative cut-off point for neonatal hypoglycemia is 2.6 mmol/L. 

Unchanged

Editors’/Reviewers’ comments-advice Specific final change made by authors Final status (page number)

 Reviewer’s advice:

Discussion section:

22. ”The current findings are comparable to others in similar settings (7-10) For instance, Sultan and his colleagues found out 73% of all babies born prematurely in South Eastern Tanzania to had been in hypoglycemic range” This statement contradict your earlier assertion that no study has been done in Tanzania (At present, there is no evidence from available/retrievable published databases of any findings on factors associated with neonatal hypoglycemia among preterm babies in Tanzania). 

 Yes, even though Sultan’s study was done in Tanzania, it referred to findings that were derived from studies performed using a cohort of all neonates and not specific to preterm babies. Besides, the study is almost two decades ago, and hence newer data were still unavailable at present. To remove ambiguity and controversies, the sentence has been rephrased to 

Rephrased in order to account for reviewer’s recommendations.

---

## [Editor Report · Decision Letter 2]

24 Mar 2024

Asymptomatic hypoglycemia among preterm newborns: a cross-sectional analysis.

PONE-D-23-33081R2

Dear Dr. Leshabari,

We’re pleased to inform you that your manuscript has been judged scientifically suitable for publication and will be formally accepted for publication once it meets all outstanding technical requirements.

Kind regards,

Kazumichi Fujioka

Academic Editor

PLOS ONE